# Negative Temperature Coefficient of Resistance in Aligned CNT Networks: Influence of the Underlying Phenomena

**DOI:** 10.3390/polym15030678

**Published:** 2023-01-29

**Authors:** Stepan V. Lomov, Iskander S. Akmanov, Qiang Liu, Qi Wu, Sergey G. Abaimov

**Affiliations:** 1Center for Petroleum Science and Engineering, Skolkovo Institute of Science and Technology, Bolshoy Boulevard 30 bld. 1, Moscow 121205, Russia; 2State Key Laboratory of Mechanics and Control of Mechanical Structures, Nanjing University of Aeronautics and Astronautics, Nanjing 210016, China

**Keywords:** carbon nanotubes, electrical conductivity, temperature coefficient of resistance, coefficient of thermal expansion

## Abstract

Temperature dependence of electrical conductivity/resistivity of CNT networks (dry or impregnated), which is characterised by a temperature coefficient of resistance (TCR), is experimentally observed to be negative, especially for the case of aligned CNT (A-CNT). The paper investigates the role of three phenomena defining the TCR, temperature dependence of the intrinsic conductivity of CNTs, of the tunnelling resistance of their contacts, and thermal expansion of the network, in the temperature range 300–400 K. A-CNT films, created by rolling down A-CNT forests of different length and described in Lee et al., Appl Phys Lett, 2015, 106: 053110, are investigated as an example. The modelling of the electrical conductivity is performed by the nodal analysis of resistance networks, coupled with the finite-element thermomechanical modelling of network thermal expansion. The calculated TCR for the film is about −0.002 1/K and is close to the experimentally observed values. Comparative analysis of the influence of the TCR defining phenomena is performed on the case of dry and impregnated films. The analysis shows that in both cases, for an A-CNT film at the studied temperature interval, the main factor affecting a network’s TCR is the TCR of the CNTs themselves. The TCR of the tunnelling contacts plays the secondary role; influence of the film thermal expansion is marginal. The prevailing impact of the intrinsic conductivity TCR on the TCR of the film is explained by long inter-contact segments of CNTs in an A-CNT network, which define the homogenised film conductivity.

## 1. Introduction

Temperature coefficient of resistance (TCR) is defined as:(1)$$\mathrm{TCR}=\frac{1}{R}\frac{dR}{dT}=-\frac{1}{G}\frac{dG}{dT};\;R=\frac{1}{G}$$
where *R* is electrical resistance (or resistivity), *G* is electrical conductance (or conductivity), and *T* is temperature.

The TCR of carbon nanotube (CNT) networks in films and nano-composites has been studied during last two decades. In the temperature range of 300–400 K, important for many applications of CNT-based sensors, negative TCRs are widely reported for random and aligned CNT configurations for practically attainable CNT volume fractions [1,2,3,4,5,6,7,8,9,10,11]. Apart from sensing, TCR may be relevant for other applications exploiting high electric conductivity of CNTs [12,13].

Three factors can affect the temperature dependence of the homogenised conductivity: (1) changes with temperature in intrinsic conductivity of CNTs, (2) changes with temperature in potential barriers at CNT contacts, influencing tunnelling conductance of these contacts, (3) configurational changes in the CNT network due to differences in coefficients of thermal expansion (CTE), influencing lengths of CNT segments between contacts and tunnelling distances. The comparative influence of these factors on the homogenised TCR is difficult to be derived from direct experiments due to the impossibility of separating involved phenomena. The problem can be approached using certain interpretations of the trends in macro-level measurements [8] or by the modelling of CNT resistive networks [9,10,14]. These modelling studies focus on the dependency of the homogenised TCR on morphological features of a CNT network, such as CNT volume fraction, orientation, and length distributions. The comparative effect of the underlying physical phenomena is investigated rather superficially. Moreover, the key parameters of the models, such as the TCR value of a single CNT, that determine the result are often chosen arbitrarily.

The present paper investigates the comparative effects for an example of CNT films, studied in [8]. As described in [8], aligned CNT (A-CNT) arrays with varying CNT length were grown in a quartz tube furnace at atmospheric pressure via a thermal catalytic chemical vapor deposition process, and re-oriented and densified using a 10 mm diameter rod and Teflon film by rolling in the alignment direction.

All properties are investigated in a temperature range of 300–400 K. The sheet resistance was evaluated using a four-point probe method (Keithley SCS-4200) where electrode–CNT connections were established using Ag paint. For the details of the films fabrication and the measurements, the reader is referred to [8].

Ref. [8] provided detailed experimental data on the sheet resistance of the films, allowing the identification of the parameters of the A-CNT geometry followed by the modelling of films’ internal structure, generating the data for the comparative study in the present paper. The A-CNT network geometry is used for the analysis of network’s TCR in the case of non-impregnated film (compared with the experimental data in [8]), and in the case of the film, impregnated with epoxy matrix.

## 2. A-CNT Films and Identification of the Nano-Structure Parameters

The parameters of multi-walled CNTs (MWCNTs), as described in [8], are shown in Table 1. The input parameters of the CNT geometry generator (maximal curvature and torsion) for similar A-CNTs were validated in [15]. These parameters are directly implemented in the modelling of our study.

As described in [8], the vertical A-CNT forest has an original thickness of *H = L*/1.5, where *L* is the CNT length. Next, it is rolled down to the thickness of ~10 µm. After this process, the film will have a fibre volume fraction of:(2)$$VF=VF_{grown}\frac{L}{1.5t}$$
where *VF_grown_* is CNT “as grown volume fraction”, and *t* is the indicative thickness of the rolled down film (see Table 1).

After the rolling, the CNTs in the film are not parallel to the substrate but are inclined, as is clearly seen in Figure 1a. The details of the resulting geometrical configuration were not investigated in [8], but they can be “reversely engineered” based on the dependency of the sheet resistivity of the films on the CNT length, given in [8].

As a simplification, we assume that after rolling, the CNTs have a prevailing orientation *α** (Figure 1b) and build a random configuration of the representative volume element (RVE), as briefly described in Appendix A, using the CNT parameters given in Table 1. The details of the geometric algorithm can be found in [15,16]. The RVE of size 1 × 1 × 1 µm is generated. This is performed for three variants of CNT length: *L =* 100, 250, and 500 µm. The corresponding CNT volume fractions, according to Equation (2), are 11%, 27%, and 53%, respectively. An example of a generated RVE is shown in Figure 1b.

The conductivity of the RVE is homogenized using the transformation of the geometrical model into a network of resistances and nodal analysis of the resulting electrical scheme, as briefly described in Appendix A. These calculations are performed for CNT electrical parameters at reference (room) temperature, taken as *T_R_ =* 300 K. In the experiments [8], two components of the anisotropic sheet resistance were measured: *R_0_* along the CNT rolling direction (*x* in Figure 1b) and *R_90_* in the cross direction (*y* in Figure 1b). For comparison with the results of calculations, the sheet resistance values can be transformed into components of the homogenized conductivity tensor as:(3)$$g_{xx}=\frac{1}{tR_{0}};\;g_{yy}=\frac{1}{tR_{90}}$$
corresponding to directions *x, y* in Figure 1b.

The conductance of the elements of the network at *T_R_* is calculated using the following formulae:Conductance of CNT connections between the contact points:
(4)$$G\left(l\right)=g_{intr}\frac{\pi D^{2}/4}{l}$$
where *D* is the CNT outer diameter (see Table 1), l is the inter-contact CNT length, and *g_intr_* is the intrinsic conductivity of the CNT.Tunnelling conductance of the CNT contact: Simmons’ formula [17,18]:
(5)$$G_{tunn}=G_{0}\frac{\tau }{s}\frac{D^{2}}{32}𝔍;\;𝔍=\exp\left(-\tau s\right)$$
where *s* is the contact distance between CNT surfaces, *s* ≥ *s_min_* = 0.34 nm, $G_{0}=2e^{2}/h$ = 7.722·10^−5^ S (*e* = 1.602·10^−19^ C is electron’s charge, *h* = 6.626·10^−34^ J·s is Plank’s constant), and the tunnelling constant:(6)$$\tau =\frac{4\pi \sqrt{2m\Delta E}}{h}$$
where *m* = 9.109·10^−31^ kg is electron’s mass and Δ*E* is the potential barrier.

For the dry network (vacuum is assumed), the tunnelling constant is about 20 nm^−1^ [19], which gives Δ*E* about 3 eV. This value is assumed for calculations (see Table 1). In the case of the absence of polymer in the tunnelling contact, there is no “polymer cutoff” effect [20]; hence, one value Δ*E* is used for any contact distance. When impregnated film is considered, the polymer cutoff distance of 0.6 nm is used, and for *s* > 0.6 nm the value Δ*E* = 2 eV is used [20].

The CNT minimal contact distance can, in principle, be affected by CNT compression. The effect of this on the homogenized conductivity was evaluated in [21] and found to be weak; therefore, the value of 0.34 nm is retained in the present calculations.

In the geometrical/electrical model, described above, there are parameters, which should be chosen to fit the experimentally determined sheet resistance:Intrinsic conductivity *g_intr_* in Equation (4), which defines the level of the homogenised conductivity and hence the sheet resistance (both its components);Angle *α**, which also affects the values *g_xx_* and *g_yy_*, but most importantly defines the ratio *g_xx_*/*g_yy_ = R*_90_/*R*_0_, which was found to be close to 1.4 in experiments [8].

In [21], the present authors compared the influence of parameters defining different conductance mechanisms and found that *g_intr_* is the most influential one and has to be fitted in the first place. When this was completed, the use of most common values for the tunnelling resistance and inter-CNT contact distance gave good correspondence with the experimental measurements in [8].

The dependency of film conductivity on CNT length, *g_ii_*(*L*), is largely defined by the change in CNT volume fraction, according to Equation (2).

The fitting process of experimental data vs. numerical predictions resulted in the parameters shown in Table 1. Figure 2 shows the results of the calculation of the film conductivity components with input parameters given in Table 1. The experimental points are calculated using Equation (3), and the sheet resistance data presented in [8]. The good correspondence with the experimentally measured values and *g_ii_*(*L*) linear trends shows that the assumptions made in the formulation of the model and the choice of model parameters were correct.

Notably, the dependence *g_ii_*(*L*) is close to linearity. The *R^2^* values of the linear fit for *g_xx_* and *g_yy_* are 0.85 and 0.87, respectively; when fitting with a power law $g_{ii}\propto L^{\alpha_{ii}}$*,* the values of *α* are equal to 0.89 for both curves and the *R^2^* values are also 0.85 and 0.87. The difference between the linear fit and the power law fit values is 2% or less (the power law fits are not presented in Figure 2 because they are nearly indistinguishable visually from the linear fits).

Hence, (see Equation (2)) the dependence *g_ii_*(*VF*) is also close to linear. Two factors, intrinsic CNT conductivity and tunnelling conductivity of CNT contacts, both influence the *g_ii_*(*VF*) dependency in different ways. For aligned CNT geometry, the input of the intrinsic CNT conductivity into the homogenised film conductivity value should be roughly proportional to *VF* (increase in the number of the parallel conductors). The tunnelling conductivity influences the homogenised one via the number of CNT contacts in the network; this number in a simplified theory of fibrous assemblies is proportional to *VF^2^* [22,23]; actually, in the aligned RVEs, which are generated in the present work, the dependency is *VF^α^*, α = 1.2 (the calculation method is the same as in [21]).

The closeness of the dependence *g_ii_*(*VF*) to linearity suggests that the prevailing influence of the intrinsic CNT conductivity on the homogenized conductivity of the film will be reflected in the relative importance of different phenomena, defining the temperature dependence of the film conductivity.

## 3. Phenomena Defining the Temperature Dependence of Resistance

The temperature dependence of material resistivity (conductivity) is characterised with temperature coefficient of resistance.

If a change over a temperature interval Δ*T = T* – *T_R_* is being considered, and derivatives in Equation (1) are changed to differences, then resistance (conductance) dependency on temperature is given by:(7)$$R\left(T\right)=R\left(T_{R}\right)\left(1+\bar{TCR\left(T\right)}\cdot \Delta T\right);\;G\left(T\right)=\frac{G\left(T_{R}\right)}{1+\bar{\mathrm{TCR}\left(T\right)}\cdot \Delta T};\;\Delta T=T-T_{R}$$
where $\bar{\mathrm{TCR}\left(T\right)}$ is effective TCR over the interval $\left(T_{R},T\right)$, $\bar{\mathrm{TCR}\left(T\right)}=\frac{\left(R\left(T\right)-R\left(T_{R}\right)\right)}{R\left(T_{R}\right)\Delta T}$.

### 3.1. Intrinsic Conductivity of CNTs

Different values for TCR of CNT intrinsic conductivity are given in the literature. For example, for MWCNTs and the temperature range 300–400 K [10,24] use the value 30 ppm/K = 3·10^−5^ 1/K (positive); Ref. [14]—the value −0.002 1/K (negative). The contradiction is resolved when physically based formulae [25,26] are used to characterise temperature dependency of intrinsic conductivity.

Refs. [25,26] define a “neutral length”, *L_N_*, as follows: if a CNT length *l = L_N_* is considered, then resistance of it is independent of temperature: “for *l < L_N_*, the number of conduction channels is the dominant parameter determining the overall resistance, and increasing temperature lowers resistance; for *l > L_N_*, electron mean free path is more important, and raising temperature increases resistance”. The neutral length is calculated as:(8)$$L_{N}=\frac{10^{3}aT_{0}D^{2}}{b+2aT_{0}D}$$
where *T*_0_ = 100 K, *a =* 2.04·10^−4^ nm^−1^K^−1^, *b* = 0.425. For *D* = 7.78 nm (Table 1), *L_N_* = 1.66 µm, which is much longer than a typical inter-contact distance in the considered CNT films, which is well below 1 µm. Therefore, in the present calculations, TCR of the CNT sections between the contacts will be negative.

The value of TCR for an MWCNT section of length *l*, using formulae in [25,26] and Equation (4), is calculated in the temperature interval 270–420 K as:(9)$$TCR_{CNT}\left(T\right)=\frac{1}{R_{*}}\frac{\pi D^{2}}{4}\frac{R_{0}}{10^{3}D}\frac{2aD+b/T_{0}}{{\left(aTD+b\right)}^{2}}\left(1-\frac{l}{L_{N}}\right)$$
where *R*_0_
*=* 1/*G*_0_
*=* 1.29·10^−4^ Ohm, *R*_∗_ = 2.5·10^5^ S/m. This value *R*_∗_ is chosen to fit the graphs shown in [25,26]. Following Equation (9), Figure 3a shows the dependency of a CNT section TCR on the section length for three levels of temperature. The dependency of TCR on *T* is weak (on the logarithmic scale): with the change in temperature from 300° K to 400° K TCR value changes by ~60%. As expected, for the section length below 1 µm, the value of TCR is negative.

Most of the inter-contact section lengths in the present models of RVE are about 0.1 µm in length; hence, TCR values will be of the order of −0.003 1/K. Note that the RVE has translational periodicity symmetry, and the inter-contact length for CNT sections crossing the RVE faces should be calculated between the contacts in two neighbouring RVEs.

### 3.2. Tunnelling Conductance of CNT Contacts

Temperature dependence of the tunnelling resistance can be calculated using the formulae in [9,10,11], which express the presence of the thermally assisted tunnelling due to the excited levels of tunnelling across the barrier [27]. The higher the temperature is, the higher the electron energy is comparative to the barrier height, generating higher probabilities to overcome it, and thereby facilitating easier tunnelling. Later, to distinguish this mechanism from others, we will refer to it as to “thermal excitation of tunnelling”. The conductance at temperature *T* will be expressed as [9,10,11]:(10)$$G_{tunn}\left(T\right)=G_{tunn}\left(T_{R}\right)\left(1+\frac{\pi^{2}}{6}{\left(\frac{k_{B}T}{\Delta E}\right)}^{2}ln𝔍\left(ln𝔍+1\right)\right)$$
where $G_{tunn}\left(T_{R}\right)$ is the conductance given by Equations (5) and (6) at the reference temperature and *k_B_* = 1.380649 × 10^−23^ m^2^ kg s^−2^ K^−1^ is Boltzmann’s constant. Differentiating Equation (10) and using Equation (1), the formula for TCR is derived:(11)$$TCR_{tunn}\left(T\right)=-\frac{2T\frac{\pi^{2}}{6}{\left(\frac{k_{B}}{\Delta E}\right)}^{2}ln𝔍\left(ln𝔍+1\right)}{1+\frac{\pi^{2}}{6}{\left(\frac{k_{B}T}{\Delta E}\right)}^{2}ln𝔍\left(ln𝔍+1\right)}$$

Figure 3b shows the dependency of thermal excitation TCR for a CNT contact on the tunnelling distance for three levels of temperature. The most often occurring contact distances are near the minimal distance of 0.34 nm [23]; hence, TCR values will be of the order of −3·10^−5^ 1/K, i.e., two orders of magnitude smaller that TCRs for CNT sections.

### 3.3. Thermal Expansion

Thermal expansion of the impregnated CNT film is calculated using Abaqus finite element (FE) modelling. The geometry of CNTs in an RVE is transferred to Abaqus. Thermo-mechanical simulation was performed using coupled temperature-displacement analysis. CNTs are represented as trusses with a stiffness of 570 GPa (calculated based on the wall stiffness of 1 TPa) and *CTE_CNT_* = 20 ppm/K [28], which are embedded in the matrix mesh. Embedded elements were widely used by the authors in their previous work on CNT composites modelling [29,30,31], as well as by others [32]. For the dry CNT film, the calculations are performed in the same manner but using zero CTE for the matrix and a very low Young’s modulus of 0.001 Pa. The boundary conditions in both cases are set according to [33].

As a result of calculations of changes in CNT network under the thermal expansion of both the CNTs and matrix, the distances between the points on CNT centrelines, corresponding to the contact positions, change from *d* in undeformed configuration to *d_TE_* after the thermal expansion. This change will be characterised by the effective thermal expansion coefficient of the inter-centreline contact distance or *CTE_d_*:(12)$$CTE_{d}=\frac{d_{TE}-d}{d\cdot \Delta T}$$

The tunnelling resistance/conductance is defined via Equations (5) and (6) by the inter-surface distance of the CNTs in the undeformed *s* and deformed *s_TE_* configuration:(13)$$s=\max\left(s_{\min},\;d-D\right)$$
(14)$$s_{TE}=max\left(s_{min},d_{TE}-D\left(1+CTE_{CNT}\Delta T\right)\right)$$
where *s_min_* = 0.34 nm is the minimal (van der Waals’) distance in between CNTs surfaces.

The corresponding effective thermal expansion coefficient of the inter-surface contact distance *CTE_s_*:(15)$$CTE_{s}=\frac{s_{TE}-s}{s\cdot \Delta T}$$

Figure 4 illustrates the calculated effective CTE of tunnelling contacts for one random realisation of the RVE, CNT length 100 µm. The behaviour in the cases of L = 250 µm and L = 500 µm are analogous.

The values of the inter-centreline distance, $CTE_{d}$, characterise the FE modelling of the thermal expansion. For the impregnated film, *CTE_d_* = 56.4 ± 9.6 ppm/K (mean and standard deviation). The film expands with the mean CTE, which is a result of homogenisation of the matrix’s and CNTs’ CTEs with variations, given by the local CNT configuration.

For the dry film, *CTE_d_* = 20.0 ± 1.1 ppm/K for all the range of inter-centreline distances, and *CTE_d_* = 20.0 ± 0.03 ppm/K for *d > D + s_min_*, the same behaviour of the “expanding film”.

In the embedded elements formulation, with the matrix mesh size larger than the inter-CNT contact distance, the nodes of the two contacting CNT may be kinematically linked to the same matrix node. This will create a numerical effect of a certain link between the CNTs, even with the very weak matrix. The CNTs in the film do not, therefore, behave as a “heap of sticks”, expanding independently. The “expanding film” behaviour can be seen as an imitation of physical phenomena of the CNT interaction caused by, for example, friction under normal forces, created by the CNT curvature and compression in the contacts [23]. Of course, this remains an heuristic explanation; full description of the contact behaviour would require a precise definition of the non-interpenetrating CNT volumes and solving the corresponding contact problems, as is performed for fibrous assemblies [34].

The transition from *CTE_d_* to *CTE_s_* can be understood if *CTE_d_* is represented for a certain contact as:(16)$$CTE_{d}=CTE_{dmean}\left(1+\varepsilon \right)$$
where *CTE_dmean_ =* 56 ppm/K for impregnated and 20 ppm/K for the dry film, where ε can be positive or negative. Evaluating ε as coefficients of variation for *CTE_d_* of dry and impregnated film, |ε|~0.2 and ~0.00015, respectively. From Equations (12)–(15), it is easy to derive for *d > D + s_min_,*
(17)$$CTE_{s}=CTE_{CNT}\left[1+\frac{d}{d-D}\left(\frac{CTE_{dmean}}{CTE_{CNT}}\left(1+\varepsilon \right)-1\right)\right]$$
which explains dependencies in *CTE_d_* shown in Figure 4b,c. Furthermore, *d < D + s_min_ CTE_s_* = 0 because of the “floor” of *s_min_* in Equations (13) and (14).

Notably for the both cases of the impregnated and the dry film, CTE’s of the tunnelling contacts are almost all positive.

The change in the tunnelling contact distance causes the change in the resistances/conductances of the contacts, characterised by the thermal expansion TCRs of contacts calculated in relation of the conductivity of the undeformed contact:(18)$${\mathrm{TCR}}_{tunn\;TE}=-\frac{G_{tunn}\left(s_{TE}\right)-G_{tunn}\left(s\right)}{G_{tunn}\left(s\right)\cdot \Delta T}$$

This change in the tunnelling conductances will cause the change in the homogenised conductivity of the film.

Figure 5 shows distributions of tunnelling contacts TCR, caused by thermal expansion for the same random RVE realisation and CNT length of 100 µm. The positive sign of TCR corresponds to the positive sign of CTE of the tunnelling contacts (Figure 4). Qualitatively, the behaviour of the dry (Figure 5a) and impregnated (Figure 5b) films are similar, with certain differences in the detail.

The majority of the contacts, ~80%, do not change their distance because of the limitation of the minimal distance between the CNT surfaces. This causes the large peaks of distributions with TCR equal 0. The width of the rest of the distribution is much larger for the impregnated film than for the dry one, corresponding to wider distributed CTEs (Figure 4b,c).

## 4. TCR of the CNT Networks and Comparative Roles of the Underlying Mechanisms

In the previous section, the temperature dependences of the resistance network elements are quantified. The nodal analysis for the conductivity homogenisation, coupled with these dependences, allows the evaluation of the comparative role of three sources for temperature dependence of the homogenised conductivity. The tunnelling conductances are under the influence of two factors: thermal excitation TCR, Equation (11); and thermal expansion TCR, Equation (14). In the coupled calculations, these two factors are assumed to influence the final TCR independently.

Such calculations were performed for randomly generated RVEs of the CNT dry films with three levels of the CNT length: 100, 250, and 500 µm, with ten random realisations per each length value.

Figure 6 shows the temperature dependence of conductivity components for dry films. There is a consistent increase in conductivity both for *g_xx_* and *g_yy_* with increase in temperature (Figure 6a). The calculated conductivity increase, *g_ii_*(*400 K*)/*g_ii_*(*300 K*), is compared with the experimental data [8] in Figure 6b. Certain differences with the experiment are seen: the conductivity increase in the experiment is the same for different CNT lengths, but in the calculations, it increases from ~1.15 for the length of 100 µm to ~1.22 for the length of 500 µm. However, the calculated TCR:TCR = (*1* − *g_ii_*(*T*)/*g_ii_*(*300 K*))/(*100 K*) = −0.0015 … −0.0022 1/K(19)
is close to the average TCR of −0.0012 1/K, as reported in [8], and is consistent with the values in the range 0.001–0.003 1/K, measured for the aligned CNT films in [2,6,7]. We attribute the difference with the experiment to that in [8]. CNTs of very high quality (defect-free) were grown to the extent that their conductance remained ballistic even at high lengths of CNT segments. On the contrary, in our calculations we assumed average quality CNTs with defects and finite intrinsic CNT conductivity (Table 1), which led to significant CNT length dependence (Figure 3a).

Table 2 shows details of the TCR calculations. It shows values calculated for dry films. The difference in the conductivity values for impregnated CNT films appears only in CTE-affected cases. It is below 0.3% and is, therefore, negligible.

Data in Table 2 give comparisons of conductivity calculated for different temperature dependence mechanisms. It shows that the temperature dependence of the intrinsic CNT conductivity plays the prevailing role in the negative TCR of the homogenised resistance. Thermal expansion gives, as could be expected, positive TCR’ thermal excitation gives negative TCR; both these contributions are one to two orders of magnitude smaller than that of the intrinsic conductivity.

This result is opposite to the conclusion by the authors of [8] who stated that the temperature dependence of the tunnelling resistance is of primary importance. This statement is not based not on direct measurements or modelling but instead on non-direct evaluations and reasoning. Moreover, Ref. [8] do not include in their reasoning thermal expansion of the network, nor the intrinsic resistance of the CNTs. The present calculations include all three factors and give simulation results that are consistent with experimental measurements, which gives credibility to our conclusions above.

Recently [35], it was demonstrated that the controversy of data on TCRs in the literature may be caused by the measurements taken from material that was not fully cured. This factor can also play a role in the experiment’s interpretation.

## 5. Conclusions

Three phenomena define TCR of CNT networks: temperature dependence of the intrinsic conductivity of CNTs, of the tunnelling resistance of their contacts, and thermal expansion of the nanocomposite. We have investigated, via a numerical modelling, the comparative role of these phenomena in the temperature range 300–400 K for aligned CNT films, created by rolling down the forests of different length, as described in [8], and the same films impregnated by epoxy matrix. The calculated TCR in the range from −0.0015 to −0.0022 1/K is close to the experimentally observed values.

For the studied materials and studied temperature interval, the main factor affecting the TCR of the A-CNT network is the TCR of the CNTs themselves. Thermal excitation TCR of the tunnelling contacts plays the second role; influence of the film thermal expansion is marginal. The prevailing impact of the intrinsic conductivity TCR in the TCR of the film is explained by long inter-contact segments of CNTs in an A-CNT network, which define the homogenised film conductivity.

## Figures and Tables

**Figure 1 polymers-15-00678-f001:**
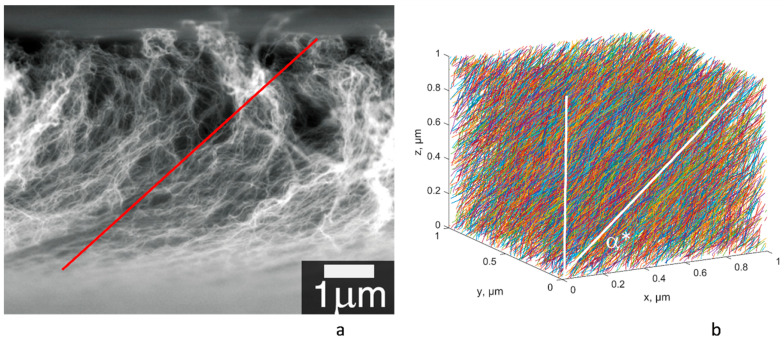
Geometry of the A-CNT film: (**a**) SEM of the film after rolling down. Reproduced with permission from Ref. [8]. Copyright 2023, AIP Publishing; (**b**) RVE of the model, CNT length 100 µm, α* = 55°. The red line in (**a**) corresponds to the assumed inclination α*.

**Figure 2 polymers-15-00678-f002:**
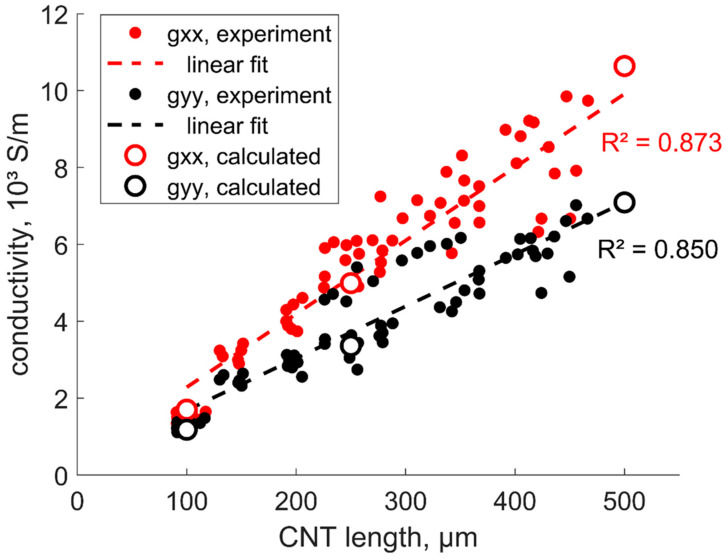
Components of the film conductivity in function of the CNT length, experiment [8] and calculations, average of results for ten random RVEs, and the size of circles corresponds to the scatter in the random simulations. R^2^ values correspond to the linear fit.

**Figure 3 polymers-15-00678-f003:**
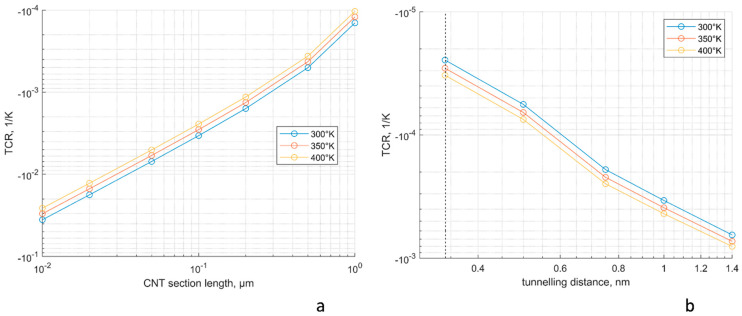
TCR for CNT sections and CNT contacts: (**a**) TCR of a CNT section, Equations (8) and (9); (**b**) TCR of a tunnelling contact due to thermal excitation; the dashed line indicates the minimum distance between the CNTs to be 0.34 nm.

**Figure 4 polymers-15-00678-f004:**
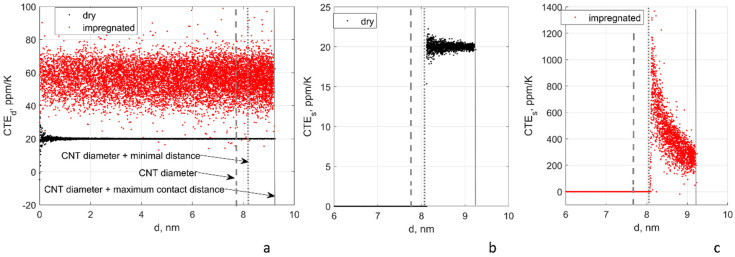
Effective CTE of the tunnelling contacts in function of the distance between the centrelines in the undeformed configuration: (**a**) $CTE_{d}$, dry, and impregnated film; (**b**,**c**) $CTE_{s}$, dry (**b**), and impregnated (**c**) film. CNT length 100 µm.

**Figure 5 polymers-15-00678-f005:**
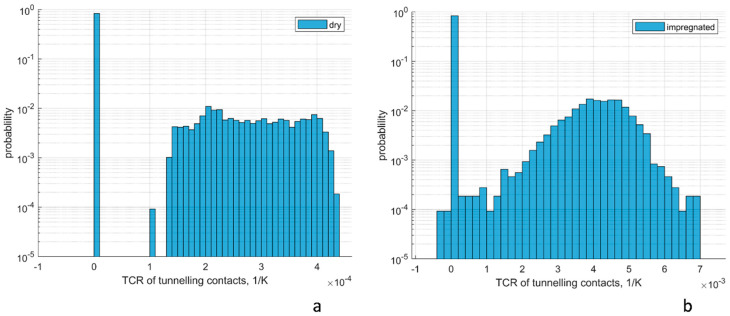
Distribution of TCR of tunnelling contacts caused by thermal expansion: (**a**) dry film; (**b**) impregnated film. CNT length 100 µm.

**Figure 6 polymers-15-00678-f006:**
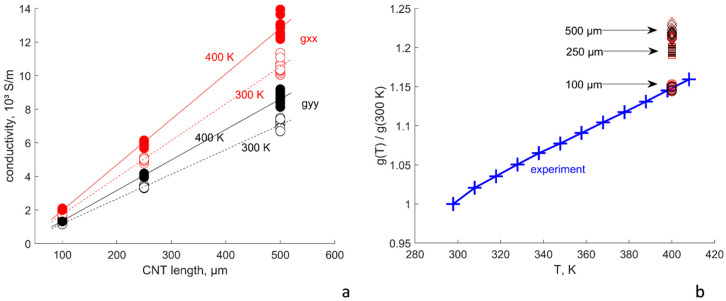
Temperature dependence of the dry film conductivity: (**a**) change in the conductivity components for different CNT length: markers—calculated values in ten random realisations, lines—linear regressions; (**b**) ratio *g(T)/g(300 K*): crosses—experimental values [8], mixed *g_xx_* and *g_yy_* components, size of the marker corresponds to the standard deviation; calculated values at 400 K: CNT length 100 µm (circles), 250 µm (squares), 500 µm (diamonds), red makers—*g_xx_*, black markers—*g_yy_*.

**Table 1 polymers-15-00678-t001:** Parameters of CNTs and the CNT film.

**Group**	**Parameter**	**Value**
Defined in [8,15]	Wall count	5
	CNT outer diameter *D*, nm	7.78
	CNT inner diameter, nm	5.12
	CNT volume fraction $VF_{grown}$, as grown	1.6%
	CNT length (*L*) range, µm	100–500
	Maximal CNT curvature *κ_max_*, 1/µm	15
	Maximal CNT torsion *τ_max_*, 1/µm	10
	Thickness of the rolled-down CNT film *t*, indicative, µm	10
Identified by fitting the resistance/length dependency in [8]	CNT intrinsic conductivity *g_intr_*, S/m	2·10^6^
	Tunnelling barrier Δ*E* in vacuum, eV	3
	Tunnelling barrier Δ*E* above the polymer cutoff distance, eV	2
	Inclination angle *α**	55°

**Table 2 polymers-15-00678-t002:** Film conductivities and TCRs, mean, and standard deviation in 10 RVE realisations, calculated for different temperature dependence mechanisms.

	CNT Length	100 µm		250 µm		500 µm	
	VF	0.11		0.27		0.53	
	Temperature Dependency Mechanism	g_xx_, 10^3^ S/m	g_yy_, 10^3^ S/m	g_xx_, 10^3^ S/m	g_yy_, 10^3^ S/m	g_xx_, 10^3^ S/m	g_yy_, 10^3^ S/m
Reference conductivity at 300 K		1.76 ± 0.05	1.18 ± 0.04	4.99 ± 0.11	3.37 ± 0.06	10.5 ± 0.41	7.07 ± 0.28
Influence of thermal effects on conductivity at 400 K	only thermal expansion	1.75 ± 0.05	1.18 ± 0.03	4.96 ± 0.11	3.35 ± 0.06	10.5 ± 0.41	7.04 ± 0.30
	only thermal excitation	1.77 ± 0.05	1.19 ± 0.04	4.99 ± 0.11	3.38 ± 0.06	10.6 ± 0.41	7.08 ± 0.28
	only intrinsic	2.03 ± 0.06	1.37 ± 0.04	5.98 ± 0.15	4.05 ± 0.07	12.9 ± 0.57	8.65 ± 0.34
	ALL	2.02 ± 0.05	1.36 ± 0.04	5.96 ± 0.15	4.03 ± 0.07	12.8 ± 0.57	8.61 ± 0.34
Influence of thermal effects on TCR over the range 300 K–400 K		TCR, 10^−3^ 1/K	TCR, 10^−3^ 1/K	TCR, 10^−3^ 1/K	TCR, 10^−3^ 1/K	TCR, 10^−3^ 1/K	TCR, 10^−3^ 1/K
	only thermal expansion	0.055 ± 0.004	0.054 ± 0.0031	0.052 ± 0.004	0.050 ± 0.0031	0.046 ± 0.005	0.050 ± 0.0063
	only thermal excitation	−0.015 ± 0.0003	−0.015 ± 0.0002	−0.015 ± 0.0002	−0.015 ± 0.0002	−0.015 ± 0.0004	−0.015 ± 0.0003
	only intrinsic	−1.53 ± 0.03	−1.52 ± 0.02	−2.00 ± 0.02	−2.01 ± 0.03	−2.22 ± 0.07	−2.23 ± 0.04
	ALL	−1.48 ± 0.03	−1.48 ± 0.02	−1.95 ± 0.03	−1.96 ± 0.03	−2.18 ± 0.07	−2.18 ± 0.04

Note: The difference of the values for dry and impregnated films is below the precision of three digits, adopted in the table, and not demonstrated.

## Data Availability

The raw/processed data required to reproduce these findings cannot be shared at this time as the data also forms part of an ongoing study.

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
