# Peer review of "Negative Temperature Coefficient of Resistance in Aligned CNT Networks: Influence of the Underlying Phenomena"

_polymers, 2023, doi:10.3390/polym15030678_

Round 1
Reviewer 1 Report
Lomov et al. demonstrated a comprehensive study of the TCR in carbon nanotubes network. especially the role of three phenomena defining the TCR: temperature dependence of the intrinsic conductivity of CNTs, the tunneling resistance of their contacts, and the thermal expansion of the network. The model is detailed and aligned well with the experimental results, making it qualified for publication after a minor revision. This referee wants to ask when the temperature goes below room T, i.e., 1.5 K-77 K, there would be more constriction. How does the model apply in that situation?
Reviewer 2 Report
- The paper makes detailed assessment of the variation of the electrical conductivity of the CNT films. Three basic mechanisms are explored in numerical simulations, with strong conclusion that only one of them makes the major contribution to experimentally observed changes. The paper is well-written and offers discussions well-supported with reference to literature, experimental evidences, and physical models. It definitely makes contribution to the state of the art and has a relevant message to deliver. - One of the questions that would deserve special attention related to the property identification procedure. Unknown constants in the derived analytical expressions, such as R* in the calculation of intrinsic conductivity, are obtained through data fitting. This is perfectly reasonable; however it raises the question whether the material constants involved in two other mechanisms could be adjusted in a similar fashion. For instance, could the effective contact distance between the CNT be tunned to fit the results? Would a different algorithm of constructing CNT model significantly affect the contact distance?
Reviewer 3 Report
This work investigates the role of three phenomena defining the TCR: temperature dependence of the intrinsic conductivity of CNTs, of the tunnelling resistance of their contacts, and thermal expansion of the network, in the temperature range 300-400 K. The analysis shows that in both cases, for an A-CNT film at the studied temperature interval, the main factor affecting network’s TCR is the TCR of the CNTs themselves. TCR of the tunnelling contacts plays the secondary role; influence of the film thermal expansion is marginal. The prevailing impact of the intrinsic conductivity TCR on the TCR of the film is explained by long inter-contact segments of CNTs in an A-CNT network, which define the homogenised film conductivity. The manuscript can be accepted for publication after some major revisions are made to address the following comments.
1. The author should provide the conductivity test details.
2. The relationship between CNT length and TCR in Fig. 3 should be deeply analyzed.
3. How do the authors prepare the CNT films?
4. Some papers related on the CNTs’ conductivity should be cited “doi.org/10.1016/j.cej.2022.137742; doi.org/10.1016/j.cej.2022.138882”.
Round 2
Reviewer 3 Report
The authors have addressed my concerns. The revised manyscript can be acceptted now.